# ULearnEnglish: An Open Ubiquitous System for Assisting in Learning English Vocabulary

**Letícia Garcia da Silva** [1], **Eduardo Gonçalves de Azevedo Neto** [1], **Rosemary Francisco** [1,*],
**Jorge Luis Victória Barbosa** [1,*], **Luis Augusto Silva** [2] and **Valderi Reis Quietinho Leithardt** [3,4]

1    Applied Computing Graduate Program, University of Vale do Rio dos Sinos, Av. Unisinos 950,
Bairro Cristo Rei, São Leopoldo 93022-750, Brazil; leeticia.garcia@gmail.com (L.G.d.S.);
eduardo7@edu.unisinos.br (E.G.d.A.N.)
2    Expert Systems and Applications Lab (ESALAB), Faculty of Science, University of Salamanca,
37008 Salamanca, Spain; luisaugustos@usal.es
3    COPELABS, University Lusófona-ULHT, 1749-024 Lisbon, Portugal; valderi@ipportalegre.pt
4    VALORIZA, Research Center for Endogenous Resources Valorization, Instituto Politécnico de Portalegre,
7300-555 Portalegre, Portugal
\*    Correspondence: rosemaryf@unisinos.br (R.F.); jbarbosa@unisinos.br (J.L.V.B.)

**Abstract:** Language learners often face communication problems when they need to express themselves and do not have the ability to do so. On the other hand, continuous advances in technology are creating new opportunities to improve second language (L2) acquisition through context-aware ubiquitous learning (CAUL) technology. Since vocabulary is the foundation of all language acquisition, this article presents ULearnEnglish, an open-source system to allow ubiquitous English learning focused on incidental vocabulary acquisition. To evaluate our proposal, 15 learners used the developed system, and 10 answered a survey based on the Technology Acceptance Model (TAM). Results indicate a favorable response to the application of incidental learning techniques in combination with the learner context. ULearnEnglish achieved an acceptance rate of 78.66% for the perception of utility, 96% for the perception of ease of use, 86.5% for user context assessment, and 88% for ubiquity. Among its main contributions, this study demonstrates a possible tool for ubiquitous use in the future in language learning; additionally, further studies can use the available resources to develop the system.

**Keywords:** English vocabulary learning; incidental vocabulary acquisition; context-aware ubiquitous learning; ubiquitous computing; open-source software

## 1. Introduction

A survey conducted by the British Council [1] showed that only 5.1% of the Brazilian population aged 16 years or older claims to have some knowledge of the English language. This proportion is considered to be a low rate, even though English as a foreign language is defined as mandatory from the sixth year of elementary school according to the standard national curriculum used in Brazilian education [2]. Second language students often fail to express their intended message precisely due to a lack of knowledge of the vocabulary of the target language. Noticing gaps in a learner's vocabulary facilitates incidental word learning from information received from the learner's environment [3].

Learning a new language is no longer limited to traditional classroom settings [4]. The increasing use of technology has changed the way English is learned and taught [5–7]. Mobile and ubiquitous technologies expand the possibilities of learning a language by allowing access in multiple contexts [8,9]. Thus, by providing access to a set of resources and tools, these technologies offer significant advantages when promoting second language learning [10].

Open-source software constitutes a strategic methodology for the collaborative development of software in different areas of knowledge. Learning environments have

explored this approach [11] in areas such as the development of technology in final degree projects [12], the improvement of learning programming [13,14], the impact of using this strategy in companies [15], the enhancement of JavaScript Simulations for learning [16], and the use of drones to support distance learning [17].

This paper proposes ULearnEnglish, a ubiquitous and open learning system to aid the teaching and learning of English language vocabulary. The scientific contribution consists of using context awareness [18] and incidental techniques for vocabulary acquisition [19]. Studies that focus on mobile device-assisted vocabulary acquisition have demonstrated effective results. Hao et al. [20] showed that students that use cell phones to study English vocabulary learn more words than when learning through other media. In addition, Chen et al. [21] confirmed that incidental vocabulary acquisition contributes to the engagement and retention of newly learned words.

In addition to developing grammar and vocabulary, language learners need to develop intercultural competence [22]: a willingness to pay attention to the cultural connections between forms, contexts, and meanings [23]. Therefore, the advantage of using ubiquitous learning is that it brings new approaches to studies, especially regarding the possibilities of personalization according to the context and profile of the learner [24].

This paper is organized into seven sections. Section 2 approaches topics related to this work, providing a background to assist in the understanding of the proposal. Section 3 presents the related scientific works to ULearnEnglish. The proposed model solution is presented in Section 5, and Section 6 presents and discusses the evaluation results. Finally, Section 7 presents the conclusions.

## 2. Background

This section aims to describe concepts related to the importance of vocabulary in learning a new language and how ubiquitous learning can assist in this context.

### 2.1. Vocabulary Acquisition Strategies

Vocabulary acquisition strategies are mainly categorized into two types: incidental and intentional [25]. Instead of learners intentionally acquiring words, incidental learning refers to the acquisition of new words in a context without necessarily looking for them [25]. Learning intentionally can reduce motivation by forcing learners to focus on specific aspects of word knowledge [19].

The knowledge of a word is built through repeated exposure to it. Users need to perceive words whose meanings they do not know from their environment; they need to become aware of and explore the relationship between words to perfect and fully develop their understanding of the words' meanings [26]. Vocabulary acquisition facilitates the development of other language skills, such as listening, speaking, reading, and writing [19].

An effective way to learn something new, mainly concerning knowledge in the long term, is learning through memory [27]. A person can increase their understanding, make relationships and associations between facts, make inferences, and reproduce information and experiences from memory. Memory is critical in teaching and learning, especially concerning a foreign language [28].

Learning strategies using memory can be essential for adult learners since frequent exposure to the language increases the learning process. Current memory techniques may involve repetition or the use of mnemonics, as Cohen et al. [28] suggested.

### 2.2. Context-Aware Ubiquitous Learning

Learning strategies and technologies have evolved together, helping each other to advance in terms of their specific knowledge and integration. In this sense, mobile and ubiquitous technologies have fostered learning strategies through the creation of mobile learning (m-learning) [29–31] and ubiquitous learning (u-learning) [32–34].

M-learning is an evolution of e-learning that allows students to carry their learning environments on their mobile devices. On the other hand, U-learning considers the integra-

tion of mobile devices, sensors, wireless communication, location/tracking mechanisms, and several other technologies to create learning environments based on context-aware computing [35] and ubiquitous computing [36]. These context-aware ubiquitous (CAUL) technologies provide a learning platform that allows continuous, more interactive, and context-aware language learning that can happen in any location or at any time [37].

Recently, ubiquitous computing has been improved with the use of time series of contexts to organize and analyze data. This new research area has been given the name of "context histories" [38–41] or "trails" [42,43]. This kind of data organization allows the exploration of advanced strategies for data analysis, such as profile management [44–48], pattern analysis [49], context prediction [50], and similarity analysis [51,52]. All these strategies allow the use of Learning Analytics [53,54]. In this scenario, ubiquitous computing and consequently ubiquitous learning encourage the collaborative development of content and knowledge [55,56], allowing a strategic platform for the use of open source software both for its development and for the development of other technological platforms based on open-source strategies.

A ubiquitous learning environment is any scenario in which the user can become immersed in the learning process [57]. For instance, while the user moves with a mobile device, the system dynamically supports the learning process through communication integrated with the environment. In this sense, cameras allow the tracking of environments to support learning [58]. Based on user preferences, the study carried out by Cohen et al. [57] showed that u-learning systems should not force a learner to follow the content established by default. The essence of u-learning is to determine which information is presented in the user's daily tasks in different forms, which is then used for learning purposes.

Research studies in u-learning have demonstrated relevant results. Mouri et al. [59] carried out a study that allowed students to record what they had learned using media and sensor data through an integrated network. This integration helped students to understand the relationships between knowledge, location, and time. Furthermore, the study of Wang et al. [60], which analyzed the use of ubiquity applied to teaching in museums, demonstrated that students have moved from being passive recipients of content to active learners using ubiquitous tools. Through such examples, it is understood that integrating systems with social and user location contexts has been effective for learning. Cardenas and Pena [9] carried out a systematic review of ubiquitous learning, allowing an overview of this area of research.

## 3. Related Works

Chen et al. [21] introduced the idea of using video games to help in vocabulary learning. They developed an adventure game with two versions: one with a game and the other with vocabulary exercises based on the game. They conducted an experiment to evaluate the game. The results showed that learners who had used the second version of the game with the vocabulary exercises retained more new words. As with ULearnEnglish, this study also applied the strategy of incidental vocabulary acquisition. However, ULearnEnglish uses the learner's day-to-day routine context to obtain new words and vocabulary, providing more opportunities to learn something new every day.

Ginn et al. [37] presented a mobile vocabulary learning application that integrates real-time object recognition and label translation. The authors used a framework consisting of three processes in recalling vocabulary: noticing a word, retrieving the word from memory, and the creative use or generative use of the word in the learner context. The study used artificial intelligence techniques, such as real-time object detection and immediate feedback. However, this study is still in the conceptual model stage.

Hao et al. [20] used cognitive learning as a pedagogical approach for English vocabulary acquisition. The authors developed a game for mobile devices (Android and smartphone) with missions that needed to be performed by students. Unlike this model, the pedagogical approach used by ULearnEnglish is the incidental vocabulary acquisition technique [19], as incidental vocabulary acquisition strategies assist in learner motivation.

Purgina et al. [23] developed a grammar learning model based on a natural language learning technique. The model allows for the configuration of the language to be learned. Teachers must perform this task by configuring the learning content in the XML format. Due to usability issues and considering the pedagogical approach, the model was developed only for tablet computers and PCs. The authors also mention the use of gamification elements to support learner engagement.

Wang et al. [8] developed a model of ubiquitous learning based on learner context. However, unlike ULearnEnglish, these works used a specific context for learning certain content. The content could be accessed by learners using their smartphone and by reading a QR code available at the learning location. In ULearnEnglish, the learning context is independent and personalized, and configurations can be made for the learning context and content. Table 1 presents a comparison of works related to the proposed model, highlighting the relevant criteria considered for the development of ULearnEnglish.

Table 1 shows that ULearnEnglish differs from other works mainly in the use of the learner context, which can be configurable and personalized—an important characteristic of ubiquitous learning. In addition, reference is made to design science research (DSR)—the adopted research method.

**Table 1.** Comparison of related works.

| Criterias | Chen et al. [21] | Ginn et al. [37] | Hao et al. [20] | Purgina et al. [23] | Wang et al. [8] | ULearnEnglish |
|---|---|---|---|---|---|---|
| Second Language Acquisition | Yes | Yes | Yes | Yes | Yes | Yes |
| Focus on learning | Vocabulary | Vocabulary | Vocabulary | Grammar | Reading and Listening | Vocabulary |
| Second language in focus | English | Customizable | English | Customizable | English | English |
| Pedagogical Approach | Incidental technique for vocabulary acquisition | Recalling vocabulary acquisition | Cognitive learning | Natural language grammar acquisition technique | Language learning for specific purposes | Incidental technique for vocabulary acquisition |
| Mobile app type | Video game native app | Native app | Native app | Native app for tablets | No details | Web-based app |
| Operating Systems | iOS | Android | Android | Android | No details | iOS and Android |
| Using the context of learner | No | Yes, object recognition | No | No | Yes, specific context | Yes, configurable context |
| Use of Gamification | Yes | Yes | Yes | Yes | Yes | No |
| Research method | Experiment | The app is a conceptual model | Quasi-experiment | Classroom experiment | Experiment | DSR |
| Audience target | College students | Apprentices in general | Apprentices in general | Teachers and students | Apprentices in general | Apprentices in general |
| Evaluated the application | Yes | No | Yes | Yes | Yes | Yes |
| TAM evaluation | No | No | No | Yes | No | Yes |

## 4. Research Method

The present study adopted design science research (DSR) [61] as the research method. This method enables the construction of a wide range of sociotechnical artifacts, such as the ubiquitous learning model proposed in this research. DSR allows researchers to

solve research problems more effectively and efficiently and to make real and practical contributions. Thus, this work followed the steps proposed by Kuechler and Vaishnavi [62] to conduct the DSR, as presented in Figure 1.

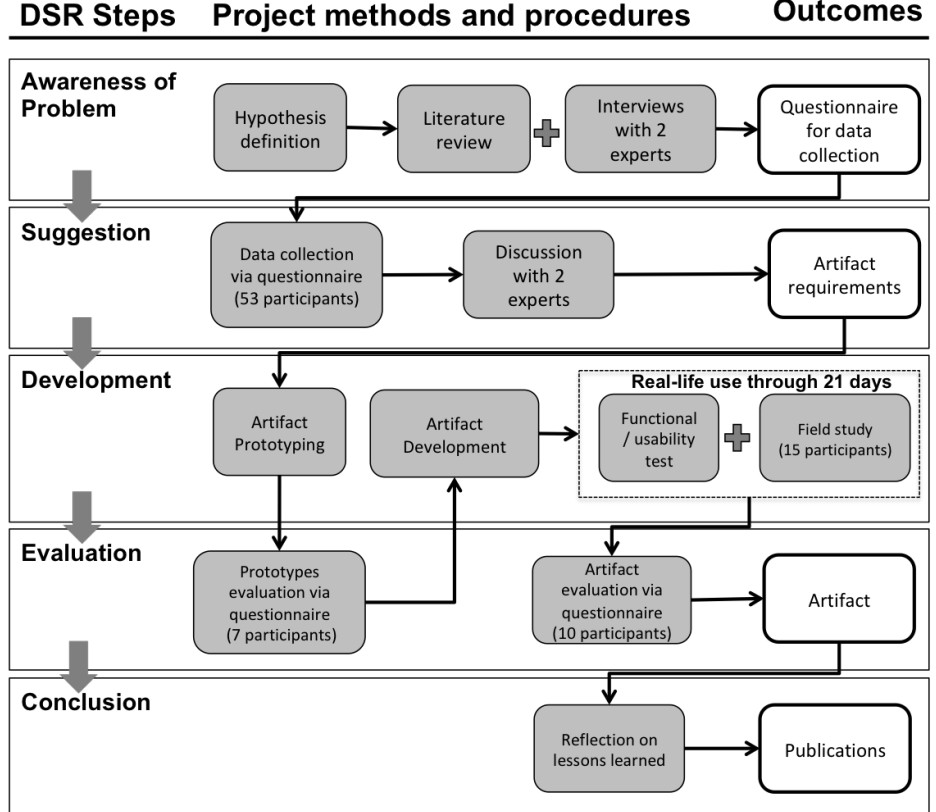

**Figure 1.** DSR steps.

The first step of the DSR consists of the determination of the research problem. This step included the following procedures in our case: (1) hypothesis definition of the research problem—that users' contexts can assist them in the learning of English vocabulary; (2) literature review, exploring the teaching approach that the application could cover, such as incidental learning and English for specific purposes; and (3) an interview with two Ph.D. professors who work as English language teachers in a university. Contact with the professors was performed more than once, which allowed us to validate the hypothesis and the elaboration of an instrument (questionnaire) to be applied in a later step.

The second step of the DSR (suggestion) consisted of applying the questionnaire developed in the first step to deepen our understanding of the problem with potential ULearnEnglish users. This data gathering allowed us to identify the real needs of users that ULearnEnglish should address. The questionnaire was operationalized through Google Forms, and 53 answers were obtained.

Based on the results of the second step, the requirements of the artifact were defined, allowing us to progress to the third step of the DSR. This step began with the prototyping of the artifact instance. The prototypes were generated for iOS and Android operating systems, and prototypes images were attached to a questionnaire for a first evaluation, mainly related to usability. In the questionnaire, respondents were asked to evaluate the screens regarding their perception of the usability and usefulness and their understanding of the application. The questionnaire was operationalized through Google Forms, obtaining responses from seven participants. After the prototypes evaluation, the development of ULearnEnglish began. Two procedures allowed the evaluation of the system: (1) the mobile application was installed and used by 15 users for 21 days; (2) from these users, 10 answered a questionnaire after trying the system.

Table 2 shows the relationship of the questionnaire statements with the evaluated aspects and presented statements elaborated based on the Technology Acceptance Model (TAM) [63]. This model was initially proposed by Davis in 1989 [64] and adapted by Chang, Yan, and Tseng [65], for the acceptance of mobile technology applied to English learning. The adapted TAM model considered the following aspects for the user to accept a given technology: perceived usefulness (PU), perceived ease of use (PEOU), perceived ubiquity value (PUV), and context (C).

**Table 2.** Statements and aspects applied to evaluate ULearnEnglish.

| | Statements | Aspect |
|---|---|---|
| 1 | The ULearnEnglish application has helped me learn and/or reinforce my understanding/study/ learning of English. | Yes |
| 2 | Repetition of content helped me memorize it more easily. | PU |
| 3 | The app's notifications encouraged me to use it more often. | PU |
| 4 | The application was easy to use without the need for help. | PU |
| 5 | I was able to use the application satisfactorily. | PEOU |
| 6 | The application correctly found locations near me. | PEOU |
| 7 | Seeing content with a location near me encouraged me to use the app more often. | C |
| 8 | The application respected my preferences when displaying content. | C |
| 9 | The use of pictures and examples helped me to memorize the content. | C |
| 10 | The information from places near me made sense, referring to the place where it was presented. | PUV |

## 5. ULearnEnglish System

This section presents the proposed system. Initially, the overview of the model is illustrated, followed by the description of the identified requirements. Finally, the details of the developed mobile application, the instance of the constructed artifact, and the aspects used to identify the user context are presented.

### 5.1. System Overview

The creation of ULearnEnglish followed the principles for the development of a CAULL (context-aware ubiquitous language learning) model. The system is intended to support the acquisition of English vocabulary according to the user's context. As a pedagogical approach, the model makes use of incidental vocabulary learning. This learning refers to the acquisition of new words in a context without the learner having necessarily looked for the words [19]. This context is observed through the resources available on the users' mobile devices.

The learning content data are stored in a relational database. The data exchange between the web server and the application is conducted using JSON (JavaScript Object Notation) format. To obtain the users' locations from their geographic coordinates, the application consumes data from the Google Places API (Application Programming Interface). Figure 2 illustrates an overview of the system and the interaction between these components.

ULearnEnglish was developed based on the MVC (Model-View-Controller) design pattern. The fundamental principle of this pattern is the division of the application into three interconnected layers to separate the presentation and user interaction with the application, the internal controls of the system logic, and the data handling [66].

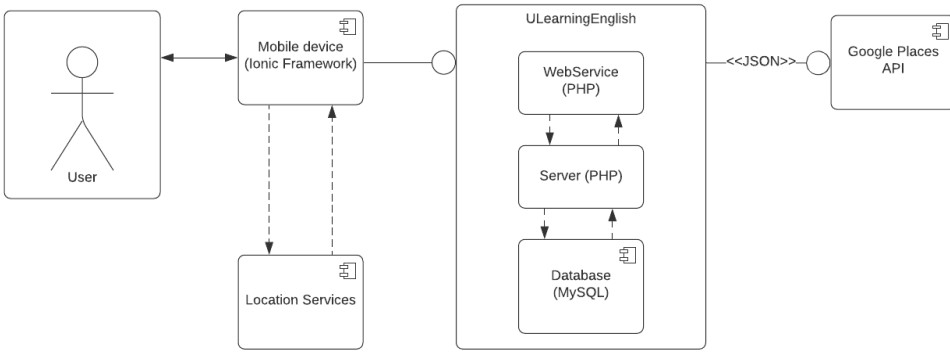

**Figure 2.** ULearnEnglish system overview.

*5.2. System Requirements*

From the results of the first and second stages of the DSR (Figure 1), the requirements that the ULearnEnglish model should cover were identified. Figure 3 shows the use case diagram with the primary requirements.

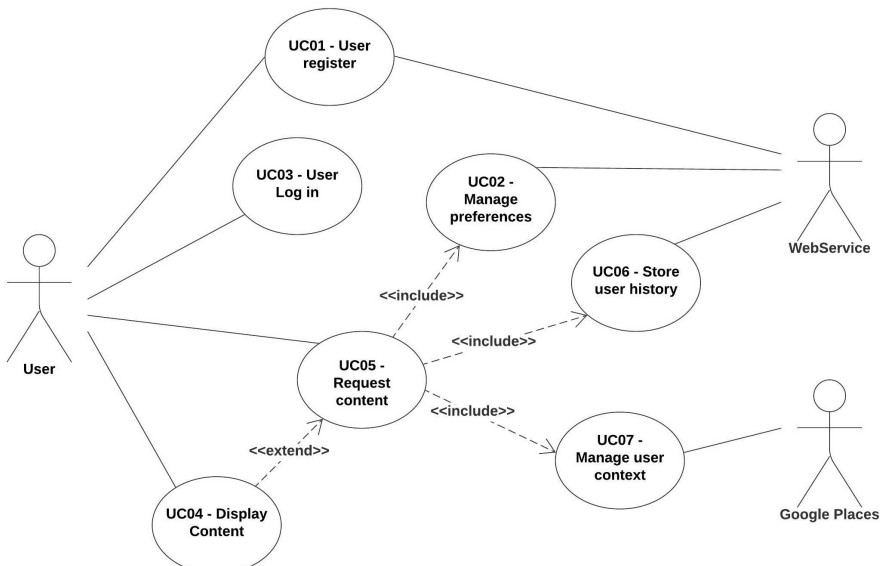

**Figure 3.** ULearnEnglish requirements.

As illustrated in Figure 3, the user must be registered and logged in (UC03) to gain access to the contents of the application. In their first contact with the system, users complete the registration (UC01) for access. After this step, users are redirected to manage their preferences (UC02), which can be changed at any time later. Once the previous steps are completed, the system obtains the user's geographic position to send the web server the content request according to the user's location and profile settings (UC07).

The web server, in turn, returns the contents to be displayed (UC04) and stores the history of the contents displayed for each location (UC06). The user has access to their brief daily history. Additionally, once the contents provided by the web server are completed, the user can request other contents (UC05).

*5.3. Implementation Aspects*

The system was instantiated and later evaluated through a mobile application developed using the Ionic framework. This framework was chosen because it allows the creation of hybrid applications (Web-based apps) for use on mobile devices with iOS or Android. Figure 4 shows three mobile application screens corresponding to the use cases UC03, UC04, and UC06 discussed in the previous section.

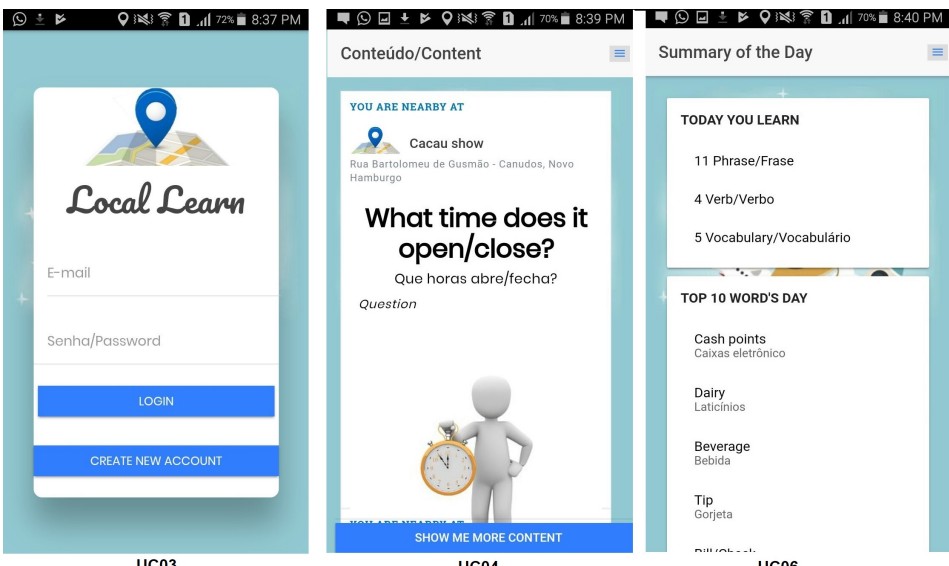

**Figure 4.** Mobile application screenshots.

For user context management, the mobile application uses Ionic's native Background Geolocation plugin as a mechanism to control the user's position. The plugin monitors a certain radius, and in case of displacement changes the geolocation variables. The user's position is passed to the Google Places API by requesting the NearBySearch method. The request returns a JSON file with information about places near the user within a parameterized radius.

For each location found in the vicinity, the JSON returns information about it. Figure 5 highlights the primary information used by the application, which are the name of the place (name), address (vicinity), and type of place (types). It is through the information contained in types that the application requests context-based content from the web-server.

```
"name" : "Academia Companhia do Corpo",
"opening_hours" : {
    "open_now" : true,
    "weekday_text" : []
},
"photos" : [
    {
],
"place_id" : "ChIJfymoa6VDGZURdcNqYMM1fxc",
"rating" : 4.6,
"reference" : "CmRRAAAAx3Mg_qCUMyrDBjedrWct-pb-CpOG15ojIzXbuqhDUjoMRMLODhBvmp6-gHylQxpv
"scope" : "GOOGLE",
"types" : [ "gym", "health", "point_of_interest", "establishment" ],
"vicinity" : "Rua Júlio de Castilhos, 246 - Centro"
,
```

**Figure 5.** NearBySearch return from a location near the user.

Table 3 shows the 19 types of places used in the evaluation. These places guided the learning content, which was obtained through the Google Places API return list. Using these places, the web service performed the database search for learning content related to the user's context. In addition, the user's preferences regarding the incidence of the content type were evaluated in terms of general vocabulary, verbs, and phrases. The types of locations were chosen by running the API in geographic coordinates tests, verifying possible returns, and by using the context diversity criterion.

Table 3. Analysis of contents by site type.

| Type of Location | Number of Contents Per Site | | | | History | | Average Appearance (5)/(4) |
|---|---|---|---|---|---|---|---|
| | Vocabulary (1) | Verbs (2) | Phrases (3) | TOTAL (4) = 1 + 2 + 3 | User Appearance (5) | % Appearance (5)/(4) | |
| Academy | 1 | 21 | 17 | 47 | 200 | 4.616 | 4.26 |
| Airport | 11 | 11 | 16 | 36 | 0 | 0.000 | 0.00 |
| Bank | 18 | 10 | 17 | 45 | 429 | 9.905 | 9.53 |
| Bar | 17 | 18 | 28 | 63 | 20 | 0.462 | 0.32 |
| Nightclub | 11 | 12 | 23 | 46 | 0 | 0.000 | 0.00 |
| Food | 19 | 16 | 22 | 57 | 541 | 12.491 | 9.49 |
| School | 19 | 20 | 30 | 69 | 441 | 10.182 | 6.39 |
| Pharmacy | 22 | 7 | 16 | 45 | 37 | 0.854 | 0.82 |
| Hospital | 18 | 12 | 20 | 50 | 231 | 5.334 | 4.62 |
| Real estate agency | 11 | 7 | 10 | 28 | 112 | 2.586 | 4.00 |
| Furniture store | 14 | 6 | 20 | 40 | 79 | 1.824 | 1.98 |
| Clothing store | 18 | 11 | 21 | 50 | 921 | 21.265 | 18.42 |
| Bakery | 19 | 13 | 16 | 48 | 8 | 0.185 | 0.17 |
| Bus stop | 4 | 15 | 12 | 31 | 0 | 0.000 | 0.00 |
| Restaurant | 24 | 18 | 24 | 66 | 540 | 12.468 | 8.18 |
| Beauty parlor | 10 | 4 | 3 | 17 | 27 | 0.623 | 1.59 |
| Health | 13 | 11 | 13 | 37 | 547 | 12.630 | 14.78 |
| Shopping mall | 25 | 14 | 34 | 73 | 28 | 0.647 | 0.38 |
| University | 23 | 21 | 29 | 73 | 170 | 3.925 | 2.33 |

## 6. Analysis and Discussion of Results

This section presents and discusses the results obtained in the ULearnEnglish evaluation.

### 6.1. Ulearnenglish Data Evaluation

Table 3 shows that 240 contents were registered and distributed in 19 types of locations; content could appear in more than one location according to its context. Among the types of locations, it was found that two had no users nearby: the airport and bus stop. Therefore, the contents registered for these locations were not displayed. The locations with the highest incidence for users were related to commerce—clothing stores—and food, as restaurants.

### 6.2. Model Acceptance Evaluation

The evaluation was conducted with participants who made use of Android devices. Participants in the second stage of the DSR were invited to use the mobile application and then answer an evaluation questionnaire. Fifteen users made use of the application, and 10 completed the questionnaire. The profiles of the participants included nine men and one woman between the ages of 16 and 35. Figure 6 presents the responses.

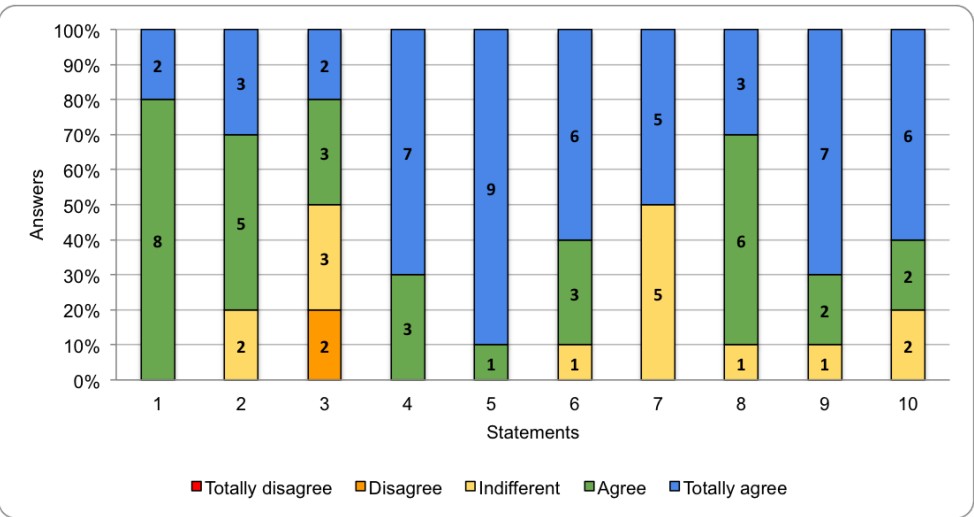

**Figure 6.** TAM evaluation results.

Following the same procedure performed by Chang, Yan, and Tseng [65], all values of the proposed scale of the statements corresponding to the aspect of TAM [65] were summed, and the result was divided by 5 (highest value of the scale) times the number of interviewees (10) times the number of questions, according to Table 4.

**Table 4.** Evaluation procedures.

| Questions | Formula | Result |
|---|---|---|
| 1,2,3 | $\sum (1,2,3) \div (5 \times 10 \times 3)$ | 0.7866 |
| 4,5 | $\sum(4,5) \div (5 \times 10 \times 2)$ | 0.9600 |
| 6,7,8,9 | $\sum(6,7,8,9) \div (5 \times 10 \times 4)$ | 0.8650 |
| 10 | $\sum(10) \div (5 \times 10 \times 1)$ | 0.8800 |

The perception of usefulness (corresponding to statements 1, 2, and 3) obtained a degree of agreement of 78.66%. The most significant divergence of opinion of users was regarding the use of notifications in the application. Users also described the usefulness aspect as "quite useful both for those who travel abroad and for those who come here" (Participant ID02) and a "very interesting idea to stimulate the learning of a new language" (Participant ID09).

The perception of ease of use (corresponding to statements 4 and 5) showed a positive acceptance index, with 96% of total agreement. This perception was also evidenced by the users' comments, which mentioned "light and easy to understand; that is, very intuitive" (Participant ID02) and "can easily be used in a practical way daily" (Participant ID09).

The evaluation of the user context (corresponding to statements 6, 7, 8, and 9) presented the result of 86.5% of total agreement. When evaluating the statements separately, it can be observed that half of the users were indifferent to learning through location; thus, it did not motivate them to increase their frequency of use.

The use of images and examples was positively accepted by most users, who also commented that "images help a lot in memorization" (Participant ID02). Some users desired to choose the type of place in which they wanted to learn at some point. Another issue that was raised was the use of sound for pronunciation. This point corresponds with the work of Sundberg and Cardoso [67], who propose the use of music for vocabulary acquisition.

Regarding motivation, one user suggested using gamification and social media interaction to encourage greater use of the application. These suggestions are in line with the findings of Hao et al. [20] and Liu et al. [68] and can be implemented in future work. It was also observed that the use of artificial intelligence in second language (L2) vocab-

ulary acquisition could be incorporated in ULearnEnglish, as presented in the study of Ginn et al. [37].

## 7. Conclusions

This study showed a positive response to context-aware ubiquitous learning (CAUL) technology to aid in learning a second language. Applying ubiquity in combination with an incidental vocabulary acquisition strategy for English vocabulary learning can engage the learner since the user's context and preferences are considered. Furthermore, using new words from learners' day-to-day routines can help learners to recall, retain, and practice the words learned, contributing to improving second language skills (reading, speaking, and listening skills).

Based on the recommendations of the DSR method, ULearnEnglish was designed with the support of specialists in the English vocabulary learning area and observing the main challenges students face when learning a new language. Besides, to evaluate its practical contribution, learners used the application in real life for 21 days. The TAM evaluation showed a favorable response to localization use to assist the participants in their English vocabulary learning. ULearnEnglish obtained acceptance rates of 78.66% for the perception of usefulness, 96% for the perception of ease of use, 86.5% for user context assessment, and 88% for ubiquity. Beyond the language teaching and learning area, this study demonstrates an opportunity for future research concerning the different areas of education in which context-aware ubiquity learning (CAUL) technology can be used to assist the learner.

Although the present study showed positive results, there are also limitations and room for improvement for ULearnEnglish. The release of the application for only one mobile platform (Android) restricted the number of users for testing. In a future study, the ULearnEnglish application should be made available for other platforms. Besides, future works could use video and audio content based on the learner context. Additionally, an integrated note-taking feature could be implemented to help learners to deepen their vocabulary comprehension. Additionally, integrated artificial intelligence techniques, such as object recognition and speech recognition, could help learners to reinforce and assess their vocabulary learning. ULearnEnglish source codes are available at a public repository and can be downloaded and developed to contribute to these future works.

**Author Contributions:** Conceptualization, L.G.d.S., E.G.d.A.N., R.F. and J.L.V.B.; investigation, L.G.d.S., R.F. and J.L.V.B.; methodology, L.G.d.S., R.F. and J.L.V.B.; software, L.G.d.S.; project administration, J.L.V.B. and R.F.; supervision, J.L.V.B.; validation, L.G.d.S., R.F. and J.L.V.B.; writing—original draft, L.G.d.S., E.G.d.A.N., R.F. and J.L.V.B.; writing—review and editing, J.L.V.B., V.R.Q.L., L.A.S. and R.F.; financial, V.R.Q.L. and L.A.S. All authors have read and agreed to the published version of the manuscript.

**Funding:** This work was supported by national funds through the Fundação para a Ciência e a Tecnologia, I.P. (Portuguese Foundation for Science and Technology) by the project UIDB/05064/2020 (VALORIZA—Research Centre for Endogenous Resource Valorization). Seed Funding ILIND–Instituto Lusófono de Investigação e Desenvolvimento, COPELABS [COFAC/ILIND/COPELABS 2020].

**Data Availability Statement:** The source codes developed in this study are available at https://github.com/legarciiaa/LocalLearn_APP (accessed on 30 May 2021).

**Acknowledgments:** The authors would like to thank the University of Vale do Rio dos Sinos (Unisinos), the Applied Computing Graduate Program (PPGCA), the Mobile Computing Laboratory (Mobilab), the Research Support Foundation of the State of Rio Grande do Sul (FAPERGS), the National Development Council Scientific and Technological (CNPq), and the Coordination for the Improvement of Higher Education Personnel—Brazil (CAPES)—Code Funding 001.

**Conflicts of Interest:** The authors declare no conflict of interest.

## Abbreviations

The following abbreviations are used in this manuscript:

| | |
|---|---|
| API | Application Programming Interface |
| CAUL | Context-aware ubiquitous language learning |
| DSR | Design science research |
| JSON | JavaScript Object Notation |
| MVC | Model-View-Controller |
| PEOU | Perceived ease of use |
| PU | Perceived usefulness |
| PUV | Perceived ubiquity value |
| TAM | Technology Acceptance Model |

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
