# Peer review of "ULearnEnglish: An Open Ubiquitous System for Assisting in Learning English Vocabulary"

_electronics, doi:10.3390/electronics10141692_

Round 1

Reviewer 1 Report

Apart from the source codes developed, the article brings forth an important issue in the field of foreign language learning, that of the development of the learners’ linguistic competence, without which the communicative competence would be severely hindered. It also offers a valuable synthesis of the applications used for language learning so far.

The study is particularly relevant especially because of the context in which it was conducted, i.e. Brazil, a country in which solutions should be readily sought for to enhance both students’ vocabulary and grammar structures and their language skills, inside and outside the classroom, either intentionally or incidentally. The solution offered by ULearnEnglish seems to be an efficient and convenient one, and it could therefore significantly help both students and language teachers (of general English and ESP) in their activities.

From our perspective, the application is worth improving and piloting with students of different language proficiency levels, from different cultural and academic backgrounds.

The suggestions for authors, concerned especially with minor language slips and rephrases, have all been included in the manuscript.

Author Response

Dear reviewer, we appreciate your criticisms and suggestions for improving our article. Attached we send a letter detailing the changes made, we also provide a pdf with all the changes in colour to better identify the changes.

Once again we appreciate your reviews.

Reviewer 2 Report

This is a wrong venue for this work. The paper is about language pedagogy or computer aided learning, where e-learning or learning is basically emphasized. This is so out of place work for this venue! Strong reject.

As for the technical merit, the quality is quite low.

Author Response

(The authors gave the same response as above.)

Reviewer 3 Report

This paper studied and describes the ULearnEnglish, an open-source system to allow ubiquitous English learning focused on incidental vocabulary acquisition and for me, the content of this paper is very informative and interesting to read. As already mentioned, the scientific contribution consists of using context awareness and incidental technique for vocabulary acquisition. The authors pointed out that focus on mobile device-assisted vocabulary acquisition have been shown to be quite effective.

From a methodical point of view, the authors developed the scientific contents on the basis of a simulations series of studies, which results are being generalised afterwards. Based on the recommendations of the DSR method, ULearnEnglish was designed with the support of specialists in the English vocabulary learning area and observing the main challenges the students face when learning a new language. This is done very convincing and provides a high grade of details, for instance: this study demonstrates an opportunity for future research concerning the different areas of education that intend to use context-aware ubiquity learning (CAUL) technology to assist the learner.

The paper is written in a very high grade of detail and especially the background (related scientific work and theoretical explanations, laboratory conditions) of this work is well researched and presented.

The overall approach and the results are widely plausible. The readability – OK

Author Response

(The authors gave the same response as above.)

Round 2

Reviewer 1 Report

The authors have made all the necessary adjustments, so I consider the manuscript ready for publication in the Electronics Journal.

Reviewer 2 Report

This still does not make sense. I checked the full scope of the journal, which is here: https://www.mdpi.com/journal/electronics/about

Such work about clear focus on language learning or pedagogy is inappropriate here. The authors write,

"The first step of the DSR consists of the awareness of the research problem. This step considered the following procedures: (1) hypothesis definition of the research problem that users‘ contexts can assist them in the learning of English vocabulary, (2) literature review, exploring the teaching approach that the application could cover, such as incidental learning and English for specific purposes, (3) interview with two Ph.D. professors who work as English language teachers in a University."

?? Very strange as this setting is nothing to deal with electronics issues as in the aims and scope of the journal.

As noted, the technical novelty is too low. Such applications have been produced for various cases in plenty. There is no real contribution to the body of knowledge in terms of software development for instance. Again, the scope of the journal clearly got into conflict with this.

Strong reject.